# Cortical Bone Loss and Fragility in a 2-Month Triple Transgenic Mouse Model of Alzheimer’s Disease

**DOI:** 10.3390/cells14221816

**Published:** 2025-11-19

**Authors:** Giuseppina Storlino, Francesca Posa, Teresa Stefania Dell’Endice, Federica Piccolo, Graziana Colaianni, Tommaso Cassano, Maria Grano, Giorgio Mori

**Affiliations:** 1Department of Clinical and Experimental Medicine, University of Foggia, Viale Pinto 1, 71122 Foggia, Italy; giuseppina.storlino@unifg.it (G.S.); francesca.posa@unifg.it (F.P.); stefania.dellendice@unifg.it (T.S.D.); piccolo.federica00@gmail.com (F.P.); 2Department of Precision and Regenerative Medicine and Ionian Area (DiMePRe-J), University of Bari Aldo Moro, Piazza Giulio Cesare 11, 70124 Bari, Italy; graziana.colaianni@uniba.it (G.C.); maria.grano@uniba.it (M.G.); 3Department of Medical and Surgical Sciences, University of Foggia, Viale Pinto 1, 71122 Foggia, Italy; tommaso.cassano@unifg.it

**Keywords:** Alzheimer’s disease, osteoporosis, 3xTg-AD mice, bone loss, bone mechanical properties, osteocytes

## Abstract

**Highlights:**

**What are the main findings?**
Characterization of the skeletal phenotype of 3xTg-AD mice with reduced cortical bone mass and decreased mechanical properties.Two-month-old male 3xTg-AD mice are characterized by an increased Number of Empty Osteocytic Lacunae and Osteoclasts, and a reduction of TRAP^+^ Osteocytes.

**What is the implication of the main findings?**
3xTg-AD mice exhibit early skeletal fragility before the appearance of typical brain lesions.This skeletal fragility could be attributed to an imbalanced process of osteocyte osteolysis.

**Abstract:**

Alzheimer’s disease (AD) and osteoporosis frequently co-occur in the elderly; however, the pathophysiological link between these two diseases remains unclear. This study investigates skeletal alterations in a triple transgenic 3xTg-AD mouse model of AD (3xTg-AD), which harbors mutations in β-amyloid precursor protein (βAPP_Swe_), presenilin-1 (PS1_M146V_), and tau_P301L,_ and recapitulates key aspects of AD pathology, including age-dependent β-amyloid plaque accumulation and cognitive decline. To assess early skeletal changes, we analyzed femurs and tibiae of 2-month-old male non-Tg and 3xTg-AD mice (*n* = 9/group) using micro-CT. Despite the absence of β-amyloid plaques at this stage, 3xTg-AD mice showed significant cortical bone loss, with reduced bone surface, periosteal and endosteal perimeters, total and cortical cross-sectional area, and polar moment of inertia. The 3-point-bending test confirmed compromised mechanical properties, including reduced maximum load-to-fracture and stiffness. Histological analyses highlighted an increased number of Empty Osteocyte Lacunae, reduced TRAP^+^ osteocytes, and an elevated number of osteoclasts; such evidence indicates impaired osteocyte function and increased bone resorption. These findings indicate that cortical bone loss and compromised mechanical properties occur before detectable neuropathological hallmarks in this AD model.

## 1. Introduction

Alzheimer’s disease (AD) is the most common age-related neurodegenerative disorder, affecting approximately 50 million people in the world [1], and with a prevalence of 15–65 per 100,000 in the age range 45–64 [2]. It is characterized by progressive and irreversible loss of major cognitive functions, such as memory, thinking, reasoning abilities, impaired speech comprehension, poor coordination, and diminished executive functions [3]. Early-stage AD patients experience compromised problem-solving, clouded judgment, sleep disorders, and depression [4]. A morphological typical feature of the disease is brain atrophy, which appears in its clinical stages and notably affects the cerebral cortex, especially in the temporo-parietal regions, hippocampus, which is involved in memory processes, and amygdala [5,6]. AD hallmarks include the accumulation of extracellular β-amyloid plaques (Aβ) and the presence of intracellular neurofibrillary tangles (NFTs) consisting of hyperphosphorylated tau aggregates [7,8]. The formation of NFTs leads to oxidative stress, producing neuroinflammation through the hyperactivation of microglia and astrocytes, and ultimately determines neuronal loss and degeneration. In addition, these impairments induce malfunction of synapses, especially in the hippocampal and cortex areas [9,10,11]. The disease reports 60–80% of all patients with manifest cognitive impairment [12]. Over the years, an increasing number of epidemiological studies have highlighted a close association between AD and osteoporosis; both degenerative diseases often coexist and share some of the risk factors such as aging, genetic aspects, female sex, and inappropriate lifestyle [12].

Osteoporosis is a metabolic disease characterized by a reduction in bone mass and strength, due to an imbalance between bone formation and resorption, with subsequent loss of bone microarchitecture and an increase in fracture risk [13]. This systemic skeletal disorder is one of the most widespread diseases in older adults; in fact, more than 10 million Americans are affected [14].

Recent reports have revealed that a considerable percentage of osteoporotic patients present a higher incidence of AD, and patients with AD show an increased incidence of osteoporosis and fracture risk [15]. Furthermore, emerging evidence has remarked that molecules involved in the onset of osteoporosis, such as Vitamin D, Osteocalcin, Osteopontin, Parathyroid hormone (PTH), and sclerostin (s) [12,13,14,15,16] may influence Aβ plaque formation, senescence, apoptosis and inflammation of nerve cells, as well as synaptic plasticity. Over time, it has become clear that there are several pathways connecting brain and bone [17]. One of them is Wnt/β-catenin, which regulates the processes of neurogenesis and synaptogenesis as well as bone formation [18,19]. Lately, it has been shown that scl, released by osteocytes (OYs) and an antagonist of the Wnt/β-catenin pathway, is involved in the onset of AD [20,21]. Some studies pointed out that scl serum levels rise with age and could be considered as an indicator of fracture risk [22,23]. Recently, it has been demonstrated that specifically scl produced by OYs leads to lower expression levels of β-catenin in the brain, increasing the expression of BACE1 (β-catenin-β-secretase1), which is involved in the formation of Aβ plaques, and reduction in cognitive performance [16]. Another pathway involved in the bone-brain axis is RANK, which is responsible for bone resorption [24]. It has been revealed that Aβ interferes with this pathway by increasing osteoclast (OC) activity [25,26].

Llabre [27] and Jung [28] independently studied skeletal alteration in the 5XFAD mouse model of AD, which features five mutations associated with AD (APPK670N/M671L (Swedish) + I716V (Florida) + V717I (London) and PS1 M146L + 286V) [29]. Jung et al. reported trabecular bone loss in 5-month-old 5XFAD mice, at a moderate stage of the disease, with cortical bone remaining unaffected at this stage, but both cortical and trabecular parameters markedly decline by 8 months, when the disease severity increased [28]. In contrast, Llabre et al. observed in 12-month-old mice shows a reduced endosteal perimeter and lower cortical bone crystallinity, a measure that indicates the size of hydroxyapatite crystals and the mineralization, without any significant change in trabecular bone [27].

The triple-transgenic (3xTg-AD) model is one of the most widely used mouse models for the in vivo study of AD and is characterized by a mutation of the PS1_M146V_, APP_Swe_, and human Tau_P301L_ genes [30,31]. The 3xTg-AD model develops Aβ plaques and NFTs in an age-dependent manner and in different brain regions, closely mimicking the development of AD in humans. These mice at two months of age fail to show the typical signs of AD, while at six months, the classic lesions of the disease begin to appear in the brain. Furthermore, Aβ deposition precedes the formation of NFTs. In particular Aβ plaques are formed initially in the cortex and then in the hippocampal region; in contrast, tau protein aggregates appear first in the hippocampal region and secondarily in the cortex. At 18 months, the brain impairment and decline in cognitive abilities result in an advanced stage [31,32]. The specific spatio-temporal appearance of these typical lesions is crucial since it mimics the development of AD pathology in the human brain [33,34,35]. To date, the characterization of brain phenotype during the progression of AD is known for this model, but only limited data are available on the bone phenotype.

In this study, we characterized the bone phenotype of 2-month-old 3xTg-AD mice, lacking the classic AD lesions in the brain, showing a loss of cortical bone mass and a reduction in the mechanical properties of bone. Interestingly, the transgenic mice displayed a higher number of empty OY lacunae in cortical bone and a lower number of TRAP^+^ OYs, indicating a decreased osteocytic osteolysis. Moreover, 3xTg-AD mice revealed an increased number and activity of OCs.

## 2. Materials and Methods

### 2.1. Animal and Experimental Design

In this study, the set of experimental procedures was carried out in strict observance of the European law implementing Directive 2010/63/EU, and the experimental protocols were examined and endorsed by the Veterinary Department of the Italian Ministry of Health (IACUC protocol number 66/2019-PR). The 2-month-old male non-Tg (*n* = 9) and 3xTg-AD (*n* = 9) mice were used for ex vivo studies. These transgenic mice were characterized by 3 mutant human genes: APP_Swe_, PS1_M146V_, and tau_p301L_. The 3xTg-AD mice were obtained from LaFerla et al. at the Department of Neurobiology and Behaviour, University of California, Irvine. The colonies of non-Tg and 3xTg-AD mice were stabularized at the University of Foggia, Italy. The strain of the 3xTg-AD mice utilized for the experiments was a C57BL6/129SvJ cross. The genotypes of the mice employed in the study were confirmed by tail genotyping. Animal housing conditions were consistent with those reported in a previous study [36].

At the end of two months of age, the mice were sacrificed, and their tissues were surgically harvested. Tibiae and femurs have been collected and properly recorded and stored. The bone segments of the femur and tibia collected at the time of sacrifice were processed according to the protocol described previously [36].All the experimental procedures were performed in compliance with standard biosafety and institutional security procedures. In addition, the investigators were blinded with regard to allotment to the group. Power analysis was conducted with a level of 0.05, yielding a sample size of *n* = 9 mice/group. The number of animals was determined based on preliminary experiments, previous related studies, and calculations performed using G*Power version 3.1.9.7. (Heinrich-Heine-Universität Düsseldorf).

### 2.2. Micro-CT Analysis

Micro-CT analysis was conducted to assess the morphological indices of both metaphyseal and diaphyseal regions of tibiae and femurs. Specifically, trabecular bone was analyzed in the metaphyseal region, excluding the growth plate, while cortical bone parameters were measured at the mid-diaphysis. The acquisition approach, image processing procedures, analysis criteria, and terminology employed were consistent with the guidelines established by Bouxsein et al. [37]. The methods used to prepare samples for micro-CT analysis have been described previously [36]. For the scanning procedure, the bone samples were positioned in the appropriate specimen holder so that their longitudinal axes were aligned with the vertical axis of the scanner, and the bones rotated around this axis during scanning. The Bruker Skyscan 1276 version 1.8 (Bruker, Kontich, Belgium) was used for image acquisition. The acquisition parameters were detailed in a previous work [36]. The reconstruction of the raw images was performed using the SkyScan software (NRecon version 1.6.10.1) (Bruker, Kontich, Belgium), obtaining 3D cross-sectional image data sets applying a 3D cone beam algorithm. During the image reconstruction phase, the following setup was applied: unified attenuation (output) range = 0.01–0.15; data were corrected for possible misalignments of overlapping sub-scans; mild beam-hardening correction = 40%; and ring artifact correction = 5. The images obtained and subsequently used were recorded in 8-bit PNG format. In aim to calibrate and calculate the volumetric bone mineral density (BMD) and tissue mineral density (TMD), a series of three hydroxyapatite (HA) phantoms (0.25 and 0.75 g·cm^−3^, and 2 mm diameter) were scanned and reconstructed using the same settings used for samples. The TC Skyscan analyzer (CTAn version 1.20.8.0) software (Bruker, Kontich, Belgium) was used to quantify the structural indices. We calculated the mean greyscale value for each phantom and constructed the curve using the appropriate algorithm available in the preferences section of the program. The same software was used to obtain indices of interest for the different samples. For cortical bone analysis of the femur, the central portion of the diaphysis was evaluated, located at the midpoint between the distal condyle and the head of the femur. The analysis considered 150 sections, starting 9 mm distal to the metaphysis. In the tibia cortical bone parameters were measured at the central portion of the diaphysis, moving 5.5 mm from the tibia condyle for a total of 200 sections (1.2 mm). Trabecular bone of the femur and tibia were analyzed in the metaphysis region, below the growth plate (1.2 and 1.9 mm, respectively), for a total of 200 sections. The cortical parameters assessed were cortical BMD, bone area (B.Ar), periosteal bone perimeter (P.Pm), endosteal bone perimeter (E.Pm), polar moment of inertia (p.MOI), and cortical bone area (B.Ar). The trabecular parameters evaluated were trabecular BMD, bone volume fraction (BV/TV), trabecular number (Tb.N), trabecular thickness (Tb.Th), and trabecular separation (Tb.Sp).

### 2.3. 3-Point Bending Test of Bone Segments

Following micro-CT analysis, femurs and tibiae were subjected to a 3-point bending test. Mechanical tests were performed using a Zwick tensile testing machine (ZwickiLine Z1.0) (sn: 734188-2019, Zwick Roell, Ulm, Germany) equipped with a 200 N load cell and the following test parameters were set: distance of lower supports: 8 mm; span length: 7.56 mm; pre-load: 0.1 N; speed until pre-load: 20 mm/min. A loading rate of 1 mm/min was applied in the medial-to-lateral direction. The Zwich/Roell testXpert III version 1.4 software was used for testing, data acquisition and analysis. The bones were cleaned of any soft tissue residue, and the fibula and peroneal bones were removed before testing. The samples were stored in 70% ethanol at 4 °C. Before mechanical testing, the samples were rehydrated in PBS and left at room temperature for at least 30 min [38]. Prior to testing, all specimen bones were measured for length and diameter at mid-shaft in the direction of breaking force using an adequate caliper. During testing, the samples were positioned similarly on the supports, with the distal end to the right, the proximal side to the left, and the posterior surface facing down. The protocol used to test, acquire and analyze data refers to the guidelines of Jepsen K.J. et al. [39].

### 2.4. Histological Analysis of Cortical and Trabecular Bone

Promptly after sacrifice, freshly dissected tibiae and femurs were instantly fixed in ice-cold 4% paraformaldehyde solution for 18 h. Then, decalcification with 0.5 M ethylenediaminetetraacetic acid (EDTA) at 4 °C was performed. Following these procedures, the bone segments were dehydrated with increasing ethanol concentration solutions and embedded in paraffin (#TC67640-Q Titolchimica, Rovigo, Italy). Paraffin blocks were cut using a standard microtome (RM-2155 Leica, Heidelberg, Germany) and stained with Hematoxylin Eosin (#HT109-500ML, #1.09844.1000 Sigma-Aldrich, St. Louis, MO, USA) and Tartrate-resistant acid phosphatase (TRAP) (#387A Sigma-Aldrich, St. Louis, MO, USA) and counterstained with Fast Green (#1.04194.0025 Sigma-Aldrich, St. Louis, MO, USA). Three sections per specimen and five high magnification fields per section were examined for analysis of OY Number, Number of Empty Lacunae, TRAP^+^ OY Number, and OCs Number. Five-micrometer-thick slices (5-μm-thick) were used for all histological analyses. To calculate the Number of Empty Lacunae, we used Hematoxylin-Eosin staining, calculated the parameter as the percentage of Empty Lacunae on the Bone Surface, and expressed it as a percentage of Empty OY Lacunae. In addition, through TRAP staining, the number of TRAP^+^ OYs was measured and expressed as the percentage of TRAP^+^ OYs for the Total Number of Lacunae (%TRAP^+^ OYs/Tot Lacunae). Again, using TRAP staining, we assessed the number of OCs and their activity by expressing them as a percentage of the Number of TRAP^+^ OCs on the Bone Surface and the percentage of Total TRAP^+^ Area on the Bone Surface, respectively. The images were viewed under a microscope Nikon Eclipse Ts2R (Nikon, Minato, Tokyo, Japan) and acquired with a 20/40x objective lens. The analysis was performed using ImageJ software version 1.53 (NIH, Bethesda, MD, USA; https://imagej.net/ij/).

### 2.5. Real Time-PCR

Total RNA was extracted from two 20 μm sections of paraffin-embedded bone tissue using the Deparaffinization Solution (#19093 Qiagen, Hilden, Germany) and the RNeasy FFPE extraction kit (#73504 Qiagen, Hilden, Germany), following the instructions provided by the manufacturer. Genomic DNA removal was performed using a kit containing the enzyme DNase I (#73504 Qiagen, Hilden, Germany). The iScript Reverse Transcription Supermix kit (#1708841 Bio-Rad, San Francisco, CA, USA) was used to complete the reversion transcription step (priming 5 min at 25 °C; reverse transcription 20 min at 46 °C, and RT inactivation 1 min at 95 °C). 1 μL of cDNA obtained was subjected to real-time PCR using the CFX Opus 96 machine (#12011319 Bio-Rad, San Francisco, CA, USA). The real-time analysis was performed following the instructions provided in the SsoFast EvaGreen Supermix kit (#172-5201 Bio-Rad, San Francisco, CA, USA) (enzyme activation 30 s at 95 °C; denaturation 5 s at 96 °C for 40 cycles; annealing 5 s at 60 °C for 40 cycles; melting curve 65–96°, with 05 °C increment every 10 s). Primer sequences are listed in Table 1 and all primers span an exon-exon junction. Two housekeeping genes, glyceraldehyde-3-phosphate dehydrogenase (*Gapdh*) and β-2-microglobulin (*β2m*), stably expressed in bone tissue, were used for normalization. Triplicates of each individual sample were tested, and relative quantification was performed by the ΔΔCt method.

### 2.6. Statistical Analysis

For the statistical analysis of sample distribution, we verified that the results obtained were consistent with a normal Gaussian distribution. Therefore, we used the Shapiro–Wilk normality test with a significance level of a = 0.05. A Welch correction *t*-test was used for parameters where values exceeded the normality test, assuming that the two groups did not have the same standard deviation (SD). Conversely, the Mann–Whitney test was used for parameters where values did not show a normal distribution. The parameters were worded as the median and interquartile range using the program GraphPad Prism 10 (GraphPad Software, Inc., La Jolla, CA, USA). For each parameter, mean values and standard errors (std.err.) are indicated in the respective figure legends. Data are displayed as box-and-whisker plots showing the median and interquartile range, from the minimum to the maximum values. All individual data points are shown. Differences were considered significant at *p* < 0.05.

## 3. Results

### 3.1. Femurs and Tibiae of 2-Month-Old 3xTg-AD Mice Showed Reduced Cortical Bone Mass, Impaired Cortical Geometry, and Resistance to Torsional Forces

To characterize the bone of 2-month-old 3xTg-AD mice, we evaluated long bones by performing micro-CT analysis on cortical and trabecular bones of the midshaft in femur and tibia (Figure 1A). The data pointed out a 1.26-fold and 1.19-fold decrease in cortical T.Ar (*p* = 0.0079) (Figure 1B) and cortical B.Ar (*p* = 0.0079) (Figure 1C), respectively, in the cortical bone of the femur in 3xTg-AD mice compared to wild-type littermates. Moreover, cortical geometry was also negatively impacted; indeed, 3xTg-AD mice showed a significant reduction of −1.16-fold in P.Pm (*p* = 0.0076) and −1.14-fold in E.Pm (*p* = 0.0042) than non-Tg mice (Figure 1D,E). Likewise, in 3xTg-AD mice, there was a decrease of −39% in p.MOI (*p* = 0.0046) compared to control mice, indicating a reduction in resistance to torsional forces (Figure 1F). At the same time, however, we did not find significant differences in the trabecular bone of the femur of 3xTg-AD compared to non-Tg mice Figure A1). The reduction in cortical bone mass, cortical geometry, and resistance to torsional forces was also featured in the tibiae (Figure 2). Micro-CT analysis revealed a decrease of 1.26-fold in both cortical T.Ar (*p* = 0.0176) (Figure 2B) and cortical B.Ar (*p* = 0.0173) (Figure 2C) in 3xTg-AD mice compared to non-Tg. Furthermore, cortical bone of Tg mice showed a significant reduction of 1.14-fold in P.Pm (*p* = 0.0002) (Figure 2D) and 1,4-fold in E.Pm (*p* < 0.0001) (Figure 2E) versus wild-type littermates. Thus, the p.MOI was also strongly diminished by 46.5% (*p* < 0.0001) in the 3xTg-AD compared to the control mice (Figure 2F).

### 3.2. The Reduction in Stiffness and Maximum Load to Fracture Characterizes 3xTg-AD Mice

Additionally, to establish the mechanical properties of the femur and tibia, we assessed the Maximum Load to Fracture and Stiffness. The analysis proved that there was a reduction in the Maximum Load to Fracture by 88% (*p* = 0.0428) (Figure 3A), a parameter indicating the greatest load a bone structure withstands before fracturing [39], in 3xTg-AD mice compared to control mice. Also, 3xTg-AD mice showed a trend of reducing Stiffness (−1.18-fold, *p* = 0.0654) (Figure 3B). This parameter indicates how much the whole bone deforms when loaded [39]. Concurrently, for the tibia, the transgenic mice exhibited a significant decrease in Stiffness by 29% (*p* = 0.0274) (Figure 3D), whereas no significant difference was observed in Maximum Load to Fracture (−1.02-fold, *p* = 0.8937) (Figure 3C) compared to non-transgenic mice.

### 3.3. 3xTg-AD Mice Showed a Higher Number of Empty OY Lacunae in Cortical Bone and a Lower Number of TRAP^+^ OYs, Indicating Reduced Osteolysis of OYs

The histological approach was used to understand which bone cells are involved in cortical bone loss. The Hematoxylin Eosin staining displayed an increased Number of Empty OY Lacunae in cortical bone, measured on the number of Total OY Lacunae, of Tg mice than non-Tg (+45%, *p* = 0.0234) (Figure 4C). This result suggests an enhanced OY apoptosis in the cortical bone of AD mice. Moreover, TRAP staining indicated a 29% decrease in the Number of TRAP^+^ OYs in 3xTg-AD mice compared to littermate control mice (*p* = 0.0361) (Figure 4D), implying that in the cortical bone of AD mice, there is an impaired osteocytic osteolysis, a process through which OYs remove and remodel their perilacunar matrix [40].

### 3.4. The Bones of 3xTg-AD Mice Exhibited an Increase in Dkk1 Expression

In order to investigate whether osteokines are involved in the development of this bone phenotype, we performed a gene expression analysis of cortical bone. Real-time PCR analysis revealed a 21-fold increase (*p* = 0.0087) in dickkopf-related protein 1 (*Dkk1*) mRNA expression levels in the bones of transgenic mice compared to wild-type littermates. This finding is noteworthy because it has been demonstrated that increased *Dkk1* is correlated with a reduction in bone mass and quality on the one hand [41,42], and, on the other hand, it is involved in facilitating both Aβ plaques formation and tau protein phosphorylation [43].

### 3.5. Cortical Bone of 3xTg-AD Mice Showed an Increased Number and Total Area of OCs

TRAP staining was performed to understand whether bone mass loss is also attributable to an enhanced number and activity of OCs. The data obtained displayed a significant increase in the OC number in cortical bone of Tg mice compared with control mice (+93%, *p* = 0.0474) (Figure 5B), while the trabecular bone of both mice groups showed an absence of significant differences (*p* = 0.4351) (Figure 5C). Accordingly, in the cortical bone of 3xTg-AD mice we also found a significant increment in the Total Area of OC activity compared with wild-type littermates (+2.3-fold, *p* = 0.0187) (Figure 5D), the evidence was not traceable in the trabecular bone of both mice groups (*p* = 0.9386) (Figure 5E).

## 4. Discussion

The 3xTg-AD mouse model is one of the models used as a gold standard for AD studies. Oddo at al. demonstrated that this model presents the occurrence of classic AD brain lesions in a spatio-temporal manner [31]. In 2019, a study conducted on 3xTg-AD mice showed that the phosphorylation of the Ser-422 site of the Tau protein, a marker of disease progression [44,45], is absent in 2-month-old mice [32], whereas the first brain lesions and cognitive decline appear at 6 months of age [30,31,32]. Our study aimed to describe the bone phenotype at the onset of any disease symptoms. Only male mice were used to avoid hormone-related skeletal changes that can occur in female C57BL/6 mice as early as 2 months of age, which could mask the disease-associated bone phenotype [46].

We are the first researchers who have focused on bone characterization in 2-month-old AD mice, showing a significant reduction in cortical bone (T.Ar and B.Ar), impaired cortical geometry (P.Pm and E.Pm), and resistance to torsional forces (p.MOI). Analysis of the trabecular bone did not reveal significant differences. Our work conducted on triple transgenic mice for AD shows how cortical bone mass reduction is already present at a very young age before manifesting the first signs of the disease.

Histological analysis conducted on the cortical bone of the femur and tibia segments revealed the cellular alteration responsible for the reduction in cortical bone mass. Interestingly, TRAP staining highlighted a greater number and activity of OCs in the cortical bone of Tg mice. According to our findings, high levels of the Receptor Activator of Nuclear Factor Kappa-B Ligand (RANKL) were found in the femur of 5XFAD mouse model, which is a sign of increased OC proliferation, differentiation, and bone resorption [28]. There are several studies showing the existence of common pathways connecting bone tissue and the brain. In particular, it has been demonstrated that the sympathetic nervous system plays a crucial role in skeletal homeostasis. That is because the adrenergic component inhibits osteoblast proliferation and function and increases the level of RANKL, the main osteoclastogenic cytokine, thereby promoting bone resorption [47,48]. Furthermore, it has been proven that the presence of Aβ plaques can increase bone resorption through the activation of osteoclastogenesis marker genes (Cathepsin K and TRAP) led by RANKL [25,26]. At the trabecular level, in contrast, TRAP highlighted no difference in the number and surface area of OCs, confirming the results previously observed by micro-CT. Moreover, the cortical bone phenotype of 3xTg-AD mice was also characterized by an increased number of Empty OY Lacunae, indicating enhanced apoptosis and OY distress. It is possible that the observed rise could be ascribed to a compromised osteocytic remodeling, a mechanism by which OYs remodel their perilacunar matrix [49]. The remodeling of OY lacunae shapes and canaliculi could free calcium from the bone matrix and also influence OY role as mechanotransducers and regulate bone turnover [40,50]. This process could potentially have a significant impact on bone tissue homeostasis. Effectively, we observed, in 2-month-old Tg mice, a reduction in the number of TRAP^+^ OYs. These cells, like mechanosensors, release molecules which play a key role in bone metabolism [20,21]. Surprisingly, gene expression analysis performed on bone samples from transgenic mice revealed increased expression levels of *Dkk1*. It is well established that *Dkk1*, by acting as an antagonist of the Wnt/β-catenin pathway in both mouse models and humans, leads to a reduction in BMD comparable to that observed in osteopenic or osteoporotic conditions [51,52]. Several studies, such as the 2019 work by Ren et al., have reported increased *Dkk1* expression in brain tissue [53]. Moreover, it has been shown that in an adult inducible *Dkk1* mouse model, in the hippocampal region, a loss of synaptic plasticity and a reduction in long-term memory is observed. This pattern seems to recover when *Dkk1* expression ends, restoring long-term memory and synapse regeneration [54]. Other studies pointed out that the excessive production of scl could be another mechanism underlying the reduction in cortical mass through the inhibition of the Wnt/β-catenin pathway, which impacts neurogenesis, synaptogenesis, and memory [55]. The use of cellular and murine models demonstrated that OYs-derived scl, through the dysregulation of the Wnt/β-catenin pathway, hurts synaptic plasticity and memory processes. Furthermore, the osteokine is involved in the mechanisms that lead to increased production of Aβ deposits, which are crucial in AD [16]. To date, the mechanisms remain unclear.

The reduction in cortical bone mass, attributable to increased OC activity, a greater Number of Empty osteocytic Lacunae (evidence of OY apoptosis) and flawed perilacunar resorption, could possibly lead to a decrease in endosteal and periosteal perimeters, and therefore to a deterioration in cortical geometry, ultimately accompanied by lower resistance to torsional forces. Our results suggest that 2-month-old Tg mice predisposed to AD exhibit reduced bone mechanical properties, including elasticity and maximum fracture load. These data are consistent with clinical studies showing that patients with AD have an improved risk of developing fractures [56,57] and, conversely, that osteoporotic patients have a higher probability of developing AD [58,59].

## 5. Conclusions

Our results demonstrate that, in 2-month-old 3xTg-AD mice, characterized by the absence of typical neurological lesions, bone loss precedes the pathogenesis of AD. Additional studies are necessary to understand how skeletal changes occur during the progression of AD.

Our study has several limitations. First, it is limited to male mice, introducing a gender bias, and focused on the pre-development stage of the disease, limiting temporal interpretation. In addition, it has a translational limitation, as both AD and osteoporosis are predominantly present in females. For this reason, future studies will need to include the characterization of the skeletal phenotype in female mice and at other stages of the disease. Moreover, subsequent studies will aim to elucidate the molecular mechanisms underlying this skeletal phenotype and to define the neuromuscular changes during the onset and progression of AD.

## Figures and Tables

**Figure 1 cells-14-01816-f001:**
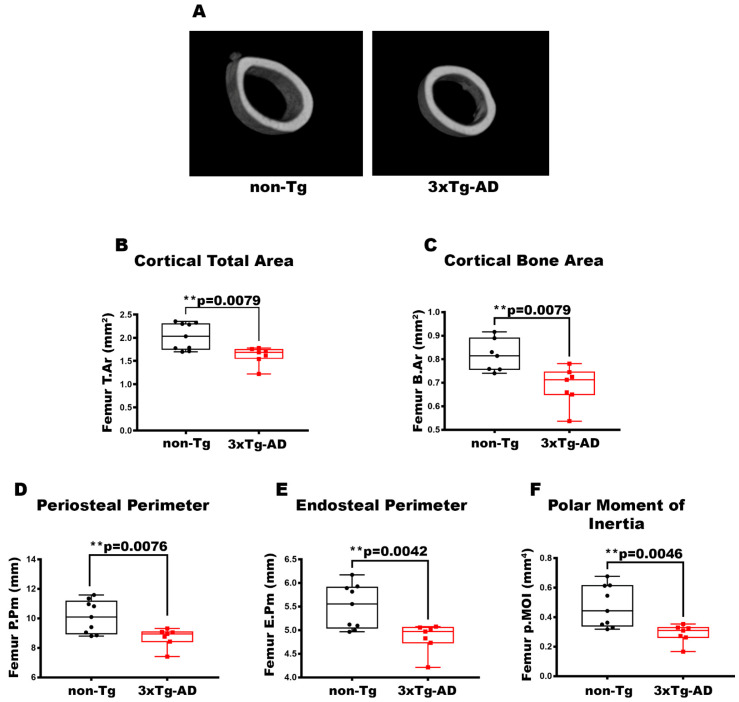
Characterization of cortical bone properties of femurs in 3xTg-AD mice. (**A**) Representative micro-CT-generated section images and calculated cortical and trabecular parameters of femurs from non-Tg and 3xTg-AD mice. (**B**–**F**) Cortical parameters included Cortical Total Area (T.Ar; mm^2^) in non-Tg (*n* = 9; mean = 2.029; std.err. = 0.0962) and 3xTg-AD (*n* = 7; mean = 1.616; std.err. = 0.073) mice; Cortical Bone Area (B.Ar; mm^2^) in non-Tg (*n* = 7; mean = 0.815; std.err. = 0.026) and 3xTg-AD (*n* = 7; mean = 0.687; std.err = 0.030) mice; Periosteal Perimeter (P.Pm; mm) in non-Tg (*n* = 9; mean = 10.11; std.err. = 0.370) and in 3xTg-AD (*n* = 7; mean = 8.71; std.err. = 0.242) mice; Endosteal Perimeter (E.Pm; mm) in non-Tg (*n* = 9; mean = 5.505; std.err. = 0.155) and in 3xTg-AD (*n* = 7; *n* = 7; mean = 4.842; std.err. = 0.115) mice and Polar Moment of Inertia (p.MOI; mm^4^) in non-Tg (*n* = 9; mean = 0.473; std.err. = 0.047) and in 3xTg-AD (*n* = 7; mean = 0.288; std.err. = 0.024) mice. The Shapiro–Wilk test was used and, based on normality, the Unpaired T test (**C**–**F**) or Mann–Whitney test (**B**) was performed. ** *p* < 0.01 vs. non-Tg.

**Figure 2 cells-14-01816-f002:**
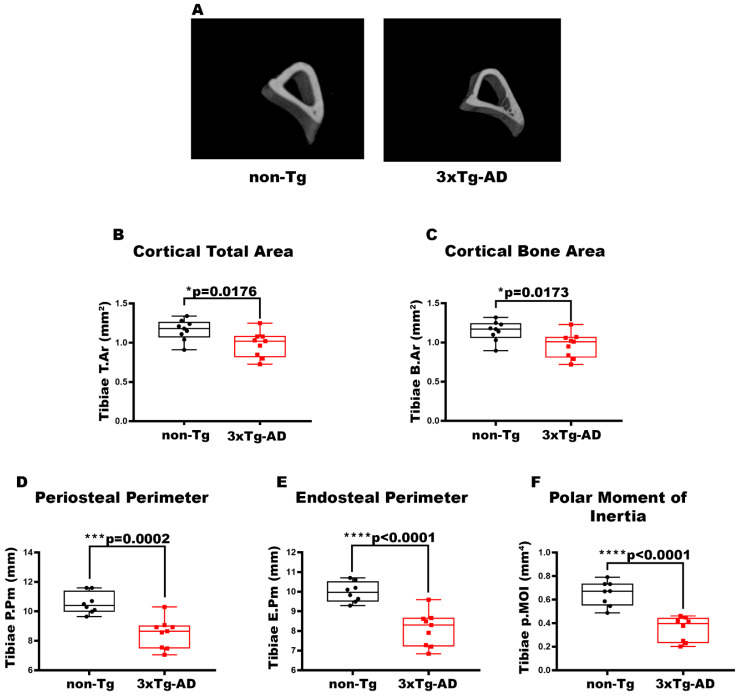
Characterization of cortical bone properties of tibiae in 3xTg-AD mice. (**A**) Representative micro-CT-generated section images and (**B**–**F**) calculated cortical parameters of tibiae from non-Tg and 3xTg-AD mice. (**B**–**F**) Cortical parameters included Cortical Total Area (T.Ar; mm^2^) in non-Tg (*n* = 9; mean = 1.162; std.err. = 0.043) and 3xTg-AD (*n* = 9; mean = 0.977; std.err. = 0.054) mice; Cortical Bone Area (B.Ar; mm^2^) in non-Tg (*n* = 9; mean = 1.146; std.err. = 0.042) and 3xTg-AD (*n* = 9; mean = 0.965; std.err. = 0.053) mice; Periosteal Perimeter (P.Pm; mm) in non-Tg (*n* = 8; mean = 10.55; std.err. = 0.255) and in 3xTg-AD (*n* = 8; mean = 8.503; std.err. = 0.333) mice; Endosteal Perimeter (E.Pm; mm) in non-Tg (*n* = 8; mean = 9.981; std.err. = 0.180) and in 3xTg-AD (*n* = 9; mean = 8.099; std.err. = 0.291) mice and Polar Moment of Inertia (p.MOI; mm^4^) in non-Tg (*n* = 8; mean = 0.652; std.err. = 0.037) and in 3xTg-AD (*n* = 8; mean = 0.351; std.err. = 0.037) mice. The Shapiro–Wilk test was used and, based on normality, the Unpaired T test (**B**–**F**) was performed. * *p* < 0.05, *** *p* < 0.001, **** *p* < 0.0001 vs. non-Tg.

**Figure 3 cells-14-01816-f003:**
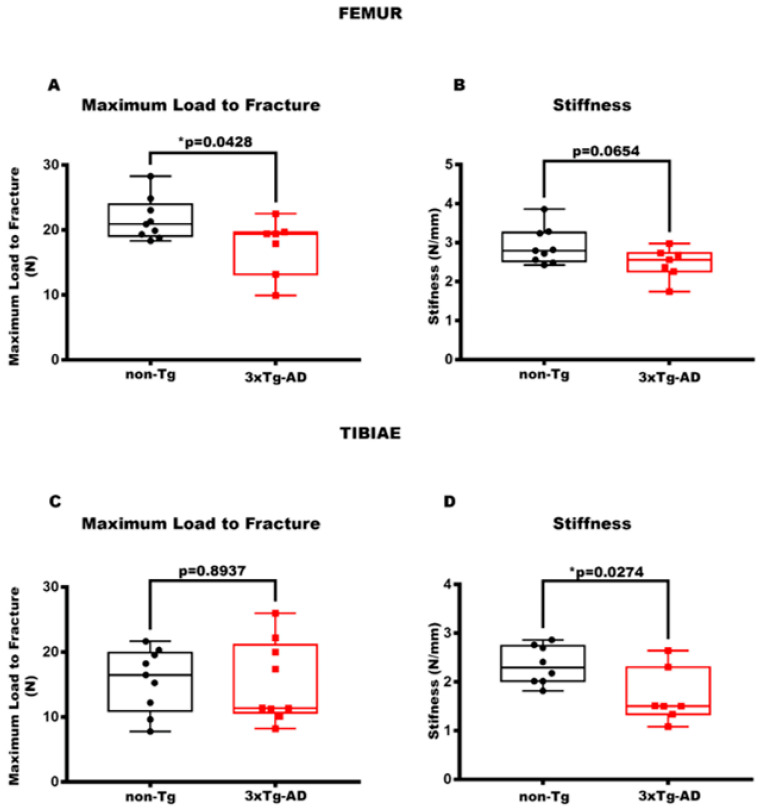
3xTg-AD mice showed a reduction in mechanical bone properties. (**A**,**B**) The parameters of the biomechanical properties for femurs included Maximum Load to Fracture (N) in non-Tg (*n* = 9, mean = 21.62; std.err. = 1.08) and in 3xTg (*n* = 7; mean = 17.40; std.err. = 1.64) mice and Stiffness (N/mm) in non-Tg (*n* = 9, mean = 2.91; std.err. = 0.15) and in 3xTg (*n* = 7; mean = 2.47; std.err. = 0.15) mice. (**C**,**D**) Also for the tibiae, the parameters considered were Maximum Load to Fracture (N) in non-Tg (*n* = 9; mean = 15.66; std.err. = 1.63) and in 3xTg (*n* = 9; mean = 15.30; sdt.err. = 2.08) mice and Stiffness (N/mm) in non-Tg (*n* = 8; mean = 2.34; std.err. = 0.14) and in 3xTg (*n* = 7, mean = 1.69; std.err. = 0.21) mice. The Shapiro–Wilk test was used, and based on normality, the Unpaired T test (**B**–**D**) was performed. * *p* < 0.05, vs. non-Tg.

**Figure 4 cells-14-01816-f004:**
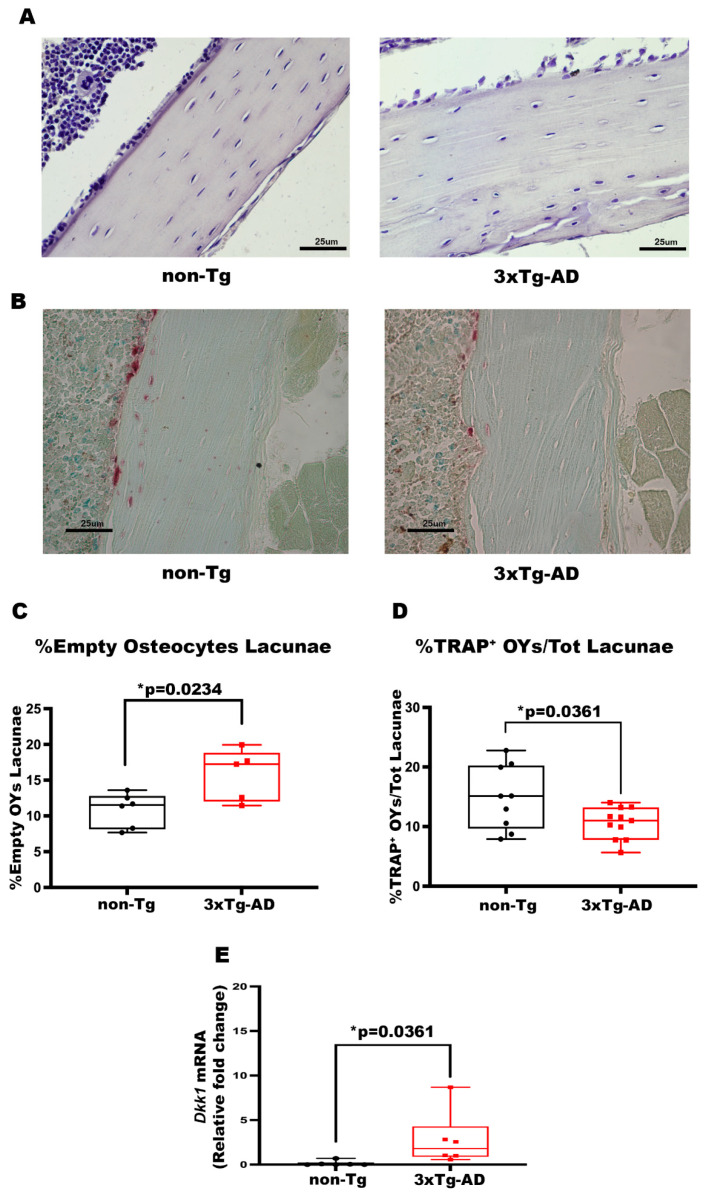
3xTg-AD mice showed an increase in the number of %Empty OY Lacunae, a reduction in %TRAP^+^ OYs and increased expression levels of *Dkk1*. (**A**,**B**) Representative images of Hematoxylin and Eosin (HE, scale bars 25 μm; magnification 20×), and Tartrate-resistant acid phosphatase (TRAP, scale bars 25 μm; magnification 20×) stains on bone sections obtained from non-Tg and 3xTg-AD mice. Histological parameters for tibiae and femurs included (**C**) Percentage of Empty Osteocyte Lacunae (%Empty OY Lacunae) obtained with HE stains in non-Tg (*n* = 6; mean = 10.85; std.err. = 0.96) and in 3xTg-AD (*n* = 5; mean = 15.78; std.err. = 1.62) mice, and (**D**) Percentage of TRAP^+^ Osteocytes/Total Lacunae (%TRAP^+^ OYs/Tot Lacunae) obtained with TRAP stain in non-Tg (*n* = 9; mean = 14.38; std.err. = 1.82) and in 3xTg-AD (*n* = 11; mean = 9.66; std.err = 1.028) mice. (**E**) mRNA expression levels of *Dkk1* (qPCR) in non-Tg (*n* = 6; mean = 0.135; std.err. = 0.112) and in 3xTg-AD mice (*n* = 6; mean = 2.77; std.err = 1.240). The Shapiro–Wilk test was used, and based on normality, the Unpaired T test (**C**,**D**) or the Mann–Whitney test (**E**) was performed. * *p* < 0.05, vs. non-Tg.

**Figure 5 cells-14-01816-f005:**
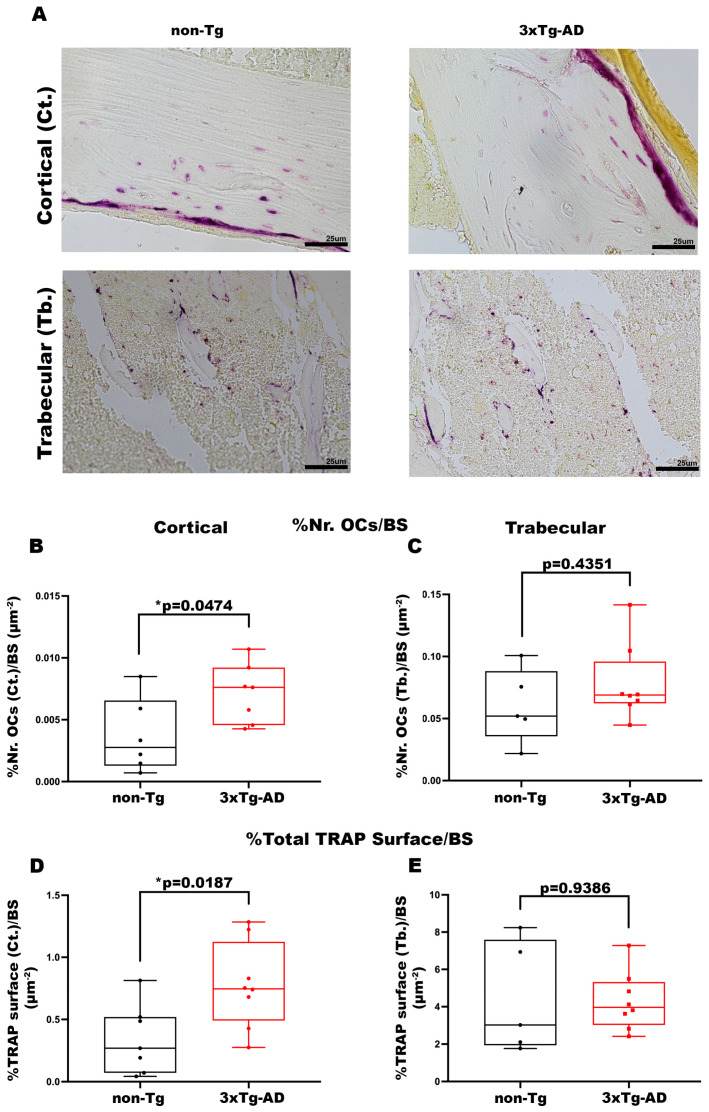
3xTg-AD mice presented an increase in the number and activity of OCs in cortical bone. (**A**) Representative images of TRAP stain on cortical and trabecular bone sections obtained from non-Tg and 3xTg-AD mice (scale bars 25 μm; magnification 20×). Histological parameters for tibiae and femurs included (**B**) Percentage of Number of OCs in Cortical Bone/Bone Surface (%Nr. OCs (Ct.)/BS) in non-Tg (*n* = 6; mean = 0.0037; std.err. = 0.0012) and 3xTg-AD (*n* = 6; mean = 0.0071: std.err. = 0.0009) mice; (**C**) Percentage of Number of OCs in Trabecular Bone/Bone Surface (%Nr. OCs (Tb.)/BS) in non-Tg (*n* = 5; mean = 0.060; std.err. = 0.013) and 3xTg-AD (*n* = 8; mean = 0.078; std.err. = 0.011) mice; (**D**) Percentage of Total Cortical TRAP Surface/BS (%Tot. TRAP Surface (Ct.)/BS) in non-Tg (*n* = 7; mean = 0.342; std.err. = 0.105) and 3xTg-AD (*n* = 8; mean = 0.777; std.err. = 0.123) mice; (**E**) Percentage of Total Trabecular TRAP Surface/BS (%Tot. TRAP Surface (Tb.)/BS) in non-Tg (*n* = 5; mean = 4.415; std.err. = 1.328) and 3xTg-AD (*n* = 8; mean = 4.299; std.err. = 0.552) mice. The Shapiro–Wilk test was used, and based on normality, the Unpaired T test (**B**,**D**,**E**) or Mann–Whitney test (**C**) was performed. * *p* < 0.05, vs. non-Tg.

**Table 1 cells-14-01816-t001:** Primer sequences. Primers were designed by using Primer Blast https://www.ncbi.nlm.nih.gov/tools/primer-blast/ (accessed on 14 September 2024).

Primer	S	AS
*Gapdh*	-acaccagtagactccacgaca	-acggcaaattcaacggcacag
*β* *2m*	-catggctcgctcggtgacc	-aatgtgaggcgggtggaactg
*Dkk1*	-aagttgaggttccgcagtcc	-gcaaacaggcaaaggtcagg

## Data Availability

The original contributions presented in this study are included in the article/Appendix A. Further inquiries can be directed to the corresponding author.

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
