# Peer review of "Cortical Bone Loss and Fragility in a 2-Month Triple Transgenic Mouse Model of Alzheimer’s Disease"

_cells, 2025, doi:10.3390/cells14221816_

Round 1

Reviewer 1 Report

Comments and Suggestions for Authors

The study addresses an important and understudied area — skeletal fragility in Alzheimer’s disease models at early pre-amyloid stages. However, the current data are purely cross-sectional, with a small, male-only cohort and lack of mechanistic or longitudinal follow-up. The discussion somewhat overstates causal inferences. Additional data addressing molecular pathways and/or time-course analyses would be required to support the strength of the claims implied in the title and abstract.

Major issues

1. The study shows early cortical bone loss in 2-month-old 3xTg-AD mice before amyloid deposition but does not provide mechanistic data linking the AD genotype to bone loss. Despite acknowledging this in the Discussion and Conclusion, the title and abstract phrase it as though bone fragility precedes and may contribute to AD pathogenesis, which may overstate the findings.
2. The study uses n = 9 mice per group, but actual n varies across measurements (e.g., in Figure 1 and Figure 3, some groups are n = 7 or n = 8). The power analysis description is minimal (stated only briefly, line 131) and no effect size or power calculation outcomes are shown. No justification is provided for attrition of samples between different analyses.
3. Only male mice were used to “avoid hormonal influences,” but: This introduces a sex bias and limits generalizability. Alzheimer’s disease and osteoporosis are both more prevalent in females. No pilot female data are provided to support excluding that sex entirely.
4. The paper focuses only on 2-month-old mice (pre-amyloid stage). No follow-up timepoints are included to track progression of bone changes alongside amyloid deposition or tau pathology. This prevents drawing conclusions about temporal sequence or mechanistic directionality.
5. The paper infers involvement of pathways like RANKL–RANK, Wnt/β-catenin, and Sclerostin based on literature but does not measure: Circulating sclerostin or osteokine levels, Expression of Wnt signaling targets, RANKL/OPG balance. As a result, the Discussion contains speculative mechanistic statements without supporting experimental evidence.
6. Although bone mechanical testing (3-point bending) is performed, neurological or behavioral assessments are absent. No correlation is made between skeletal findings and AD-related biomarkers or brain pathology. Therefore, the link between early skeletal fragility and AD pathogenesis remains indirect.

7.Lack of mechanistic data (e.g., no measurements of Wnt signaling, sclerostin, or RANKL pathways despite mechanistic claims).
8. Cross-sectional design at a single time point, insufficient to support the temporal inference implied by the title (“precedes”).
9. Small sample size with variability between figures and male-only animals, limiting robustness and generalizability.
10. Overstated causal language in the title and abstract, inconsistent with the descriptive nature of the data.
Minor issues
1. Statistical methods are described briefly, but: There is no correction for multiple comparisons, despite many endpoints. Normality testing is mentioned, but actual distributions are not shown. p-values are sometimes presented with inconsistent formatting. p < 0.5 is likely a typo in line 228 and should be p < 0.05.
2. Some terms are vague or imprecise, e.g.: “robust reduction” (line 236), “strong reduction” (line 278), “profound decrease” (line 246), which should be replaced with quantitative language.
3. The title may imply causality (“Precedes”) which the study does not fully prove.
4. Some figures have small sample sizes (n=5–7) for histology, which reduces statistical reliability.
5. Boxplots are used but effect sizes are not reported.
6. Scale bars and magnifications in some histological figure legends are inconsistent or incomplete.
7. Micro-CT scan parameter table is present but BMD calibration details are minimal.
8. The background discussion is somewhat long and repetitive, especially in the introduction, with multiple paragraphs reiterating bone-brain axis mechanisms. This could be more focused.
9. The authors acknowledge lack of mechanistic exploration and muscle assessment (line 416), but: No mention of sex bias, lack of longitudinal data, or small sample size in the limitations. No reflection on translational relevance vs. model limitations.

Reviewer 2 Report

Comments and Suggestions for Authors

This study investigated 2-month old 3XTg AD model for the their bone phenotype. This is the age before neurological phenotype for AD appeared and revealed severe bone cortical bone deterioration but not trabecular bone.  The mechanism was determined due to the decreased TRAP in osteocytes and increased TRAP in osteoclasts in the cortical bone. Overall, the study was well done and data are presented clearly. Following are some of the comments.

  1. How to identify empty OY lacuna?
  2. Line 116, “Obtained by” should be “Obtained from”.
  3. Line 120, “Tail genotyping” need more details, primers and PCR conditions should be provided.
  4. Line 135-136. Why authors measure metaphysis while they are saying later they measure cortical bone? This area is rich in trabecular bone. Usually, we measure diaphysis for cortical bone. I guess here should the entire tibia and femur were scanned as later on in this paragraph, authors said midshaft femur cortical bone.
  5. Line 139, “The bones were placed in specimen” Should be “Specimen holder”?
  6. Line 163 to 164, Proximal tibia metaphysis trabecular bone should not include area from proximal to distal growth plate as growth plate is cartilage. The VOI should be below growth plate. This should be clarified as later authors said distal of growth plate.
  7. Line 201, “Include in paraffin” meaning is not clear. It should be “embedded in paraffin”.
  8. Line 206, “High-power field” may be changed to high magnification.
  9. Line 228, Significant difference at “P<0.5” ? This is usually P<0.05.
  10. Line 236, “Figure 1” Should be “Figure 1A”.
  11. Line 238: “Compared to littermate”, what Littermate? “WT littermate”?
  12. Line 276: 3.2 Subheading should be rephrased.
  13. Figures1 and 2 microCT images appeared not in the midshaft from the shape of the cortical bone, these areas still have trabecular bone. In the midshaft femur, usually you can not see the trabecular bone.
  14. Can authors show the body weight of these 3XTG mice? Cortical bone and biomechanical properties correlated with cortical bone thickness and body size.
  15. Can authors also show Proximal tibia (metaphysis) trabecular bone parameters? Even though they have no differences, they are the results the authors analyzed. At lease this can be out in the supplemental data. In many cases, trabecular bone is more sensitive to bone loss.
  16. Line 303, What Littermate? Please clarify.

  1. Figures 4 and 5: TRAP+ OYs, TRAP+ is an osteoclast marker, it supposes not express in osteocytes, only expressed on bone surface osteoclasts. But authors showed positive staining in osteocytes. Using Sigma 387Kits for TRAP, Hematoxylin counterstaining should be performed to identify tissues or cells types. At least this should be done for future direction.
  2. Figure 4A, Cortial and trabecular bone label should be like the Y axis of histogram.

            It is hard to read in the current way. 

  1. Since Wnt/beta-catenin signaling may play a role in the brain-bone axis, author should examine this siginaling pathway in the endosteal and periosteal osteoblasts or osteocytes.
  2. Line 377, “Bunch” should be replaced with a scientific word, such as “many” or “several”.

Comments on the Quality of English Language

Minor English corrections should be made. 

Reviewer 3 Report

Comments and Suggestions for Authors

The hallmark of AD is the extracellular deposition of Ab plaques. On the other hand, osteoporosis is characterized by reduction in bone mass and strength. Recent reports have shown that a considerable percentage of osteoporotic patients have a higher incidence for AD, and patients with AD have higher incidence for osteoporosis. This indicates that there may be a connection between these diseases, but functional association is missing.

Here, the authors have used the 3xTg-AD mouse model of AD to study the development of osteoporosis. The study was carried out in 2 months old 3xTg-AD mice. At this age, the 3xTg-AD mice have not developed Ab deposits.

They found that the 2 months old 3xTg-AD mice had increased number and activity of osteoclasts as well as certain alterations in their bone structure (tibiae and femurs) with micro-CT.

The study is well-designed and supports the existence of a brain-bone axis.

There are only the following minor points that need to be address.

  1. The rational for using male animals is not convincing, Usually, females are more prone to osteoporosis and should have been studied. Lines 350-352 contain a description that is not clear. Why hormonal-related changes will mask any skeletal phenotype and not worsen it? In contrast the latter will make it easier. This should be clarified in detail.
  2. It appears that an analogous study has been carried out in 5XFAD mice (line 363-364). The novelty of this work is the fact that the authors demonstrated that lesions started at the age of 2 months before Ab deposits. This description on 5XFAD should be transferred from Discussion to the Introduction.
  3. Line 383 Catepsin K is probably Cathepsin K

Round 2

Reviewer 1 Report

Comments and Suggestions for Authors

The original manuscript was descriptive but written as if it were mechanistic and causal, with design and reporting gaps too large to meet publication standards at that stage. The revised version demonstrates substantial improvement in scientific accuracy, transparency, and presentation. While the study remains descriptive, it now honestly presents itself as such — with limited mechanistic exploration but a coherent, defendable scope.

1. The phrase “Tg mice predisposed to AD exhibit greater susceptibility to fractures” remains slightly causal in tone.
2. No histological or immunostaining correlation (e.g., for Dkk1 or osteocyte apoptosis) to reinforce molecular data.
3. In the statistical analysis, no correction for multiple comparisons (e.g., Bonferroni or Holm). Effect sizes (Cohen’s d, η², etc.) are not reported, though the sample size is small. Normality testing is mentioned, but distribution plots or residual analyses are not shown.
4. Three-point bending test parameters (span length, loading rate, sample hydration status) are not specified.
5. Conclusion repeats statements from the Discussion and lacks a forward-looking remark (e.g., specific next steps).
6. Minor stylistic inconsistencies (spacing, capitalization of gene names, e.g., “Scl” vs. “sclerostin”).
7. Reference formatting could be harmonized (some use abbreviated journal names inconsistently).

Round 3

Reviewer 1 Report

Comments and Suggestions for Authors

Thank you for submitting the revised version of your manuscript and for providing a detailed response to the reviewer comments. I appreciate the considerable improvements made in statistical analysis, reporting clarity, and overall structure of the manuscript. The addition of multiple-comparison corrections, effect sizes, QQ plots, and expanded methodological details significantly strengthens the scientific rigor of the study.

After reviewing both your responses and the revised manuscript, I find that several issues have been satisfactorily addressed. However, one of the major concerns from the previous review remains only partially resolved and requires further revision before publication.

1. Although you have revised some sentences to adopt a more descriptive tone, the Discussion section still contains multiple statements that imply causal or mechanistic explanations that are not directly supported by the experimental evidence presented. Examples include assertions regarding:
(1) “Histological analysis … revealed the cellular alteration responsible for the reduction in cortical bone mass.” This implies responsible for, a mechanistic conclusion. But histology is limited (mostly TRAP staining), and no apoptosis markers or osteoblast/osteocyte functional assays were included.
(2) “The osteoclastic cytokine RANKL… is a sign of increased OC proliferation, differentiation, and bone resorption.” This is mechanistic and fits neither the modest dataset nor the reviewer’s request for immunostaining (which they did not add).
(3) “This process could potentially have a significant impact on bone tissue homeostasis.” Mechanistic speculation remains too strong unless justified with more cautious language ("may suggest", "is consistent with").
(4) “Another mechanism underlying the reduction… is the inhibition of the Wnt/β-catenin pathway…”
This is highly mechanistic but not shown in their study.

Add one to two sentences in the Limitations subsection explicitly stating that mechanistic interpretations remain hypothetical due to the absence of immunostaining, apoptosis markers, or protein-level validation.

2. Some stylistic inconsistencies remain (capitalization, gene/protein formatting, minor spacing issues). Please ensure that all gene names follow standard formatting throughout the text and that reference formatting conforms uniformly to journal guidelines.